# Chemistry in Fungal Bioluminescence: Theoretical Studies on Biosynthesis of Luciferin from Caffeic Acid and Regeneration of Caffeic Acid from Oxidized Luciferin

**DOI:** 10.3390/jof9030369

**Published:** 2023-03-18

**Authors:** Xiayu Liu, Mingyu Wang, Yajun Liu

**Affiliations:** 1Key Laboratory of Theoretical and Computational Photochemistry Ministry of Education, College of Chemistry Beijing Normal University, Beijing 100875, China; 2School of Science, Hainan University, Haikou 570228, China; 3Center for Advanced Materials Research, Beijing Normal University, Zhuhai 519087, China

**Keywords:** fungal bioluminescence, DFT, caffeic acid cycle, luciferin

## Abstract

Fungal bioluminescence is widely distributed in the terrestrial environment. At a specific stage of growth, luminescent fungi shine green light at the fruiting body or mycelium. From the viewpoint of chemistry, fungal bioluminescence involves an in vivo cycle of caffeic acid. The complete cycle is composed of three stages: biosynthesis of luciferin from caffeic acid, luminescence process from luciferin to oxidized luciferin, and regeneration of caffeic acid from oxidized luciferin. Experimental studies roughly proposed this cycle but not the detailed reaction process and mechanism. Our previous theoretical study clearly described the mechanism of the middle stage. The present article attempts to describe the reaction processes and mechanisms of the other two stages by theoretical calculations. A complete theoretical study on the chemistry in the entire process of fungal bioluminescence is helpful to deeply understand fungal bioluminescence.

## 1. Introduction

Bioluminescence (BL) is a natural phenomenon of visible light emission from living organisms [1]. In most cases, BL means a series of chemical reactions starting from luciferin catalyzed by luciferase, which converts chemical energy to light [2]. Of thousands of species in the kingdom Fungi, approximately 100 species have been identified as bioluminescent [3,4]. All bioluminescent fungi continuously emit similar green light (~530 nm) during their steady period of life [3,5], and the light attracts insects to help spores spread in a windless environment [6]. In 2018, through a breakthrough study, Kotlobay’s group reported a caffeic acid circulating process of fungal BL (see Figure 1) [4]. As the raw material for the biosynthesis of luciferin, caffeic acid widely exists in the metabolic process of plants. In 2020, Mitiouchikina’s group developed a self-luminous tabacum named *Nicotiana tabacum* and *Nicotiana benthamiana* [7], based on the fungal BL system by integrating related genes into the plant nuclear genome. This kind of tabacum can emit light at all development stages, and the luminous intensity is much higher than that of luminous plants based on the bacterial BL system [8]. Recently, Khakhar et al. found that inserting the same genes into several other plant species, including tomatoes and dahlias, can also produce spontaneous luminescence [9].

Previous research has indicated that all luminescent fungi share the same BL system [10]. As shown in Figure 1, the caffeic acid cycle in the fungal BL system consists of three stages. In Stage 1, caffeic acid is used as raw material to produce fungal luciferin through biosynthesis. This stage can be divided into two steps: (1) hispidin synthase (HispS) utilizes caffeic acid to synthesize hispidin, and (2) under the catalysis of hispidin 3-hydroxylase (H3H), hispidin is hydroxylated and converted into 3-hydroxyhispidin, which is fungal luciferin. In Stage 2, luciferin is catalyzed into fungal luciferase (Luz) and is oxidized to produce caffeoyl pyruvate (fungal oxidized luciferin) and emit light. In Stage 3, oxidative luciferin is metabolized into pyruvic acid and caffeic acid in caffeoyl pyruvate hydrolase (CPH).

Although the general reaction process of the caffeic acid cycle has been reported, the details and mechanism of each reaction step remain unknown. The crystal structures of the related four key enzymes in fungal BL have not been revealed [4], so it is impossible to investigate the reaction mechanism in the protein environment. At present, we can only describe the possible reaction process and mechanism by theoretical simulation in a non-protein environment based on sporadic experimental evidence. Among the three stages of fungal BL, Stage 2 has been theoretically studied by our group, and the related details and mechanism of related reactions have been clearly described [3]. In this article, we employed density functional theory (DFT) to uncover the mechanism of the other two stages. We aim to draw a complete picture for understanding the entire process of fungal BL.

## 2. Computational Model and Method

### 2.1. Computational Model of Stage 1

As shown in Figure 1, several molecules are involved in Stage 1. During the biosynthesis of hispidin (Step 1), the combination of caffeic acid and CoA in the presence of the adenosine triphosphate–magnesium ion complex (Mg-ATP) takes place. After this process, caffeoyl-CoA achieves the growth of the carbon chain through a condensation reaction, which involves the process of transferring caffeoyl from CoA to cysteine. During the biosynthesis of luciferin (Step 2), flavin in the biological system is converted into the intermediate of flavin peroxide through a series of reactions, and participates in the hydroxylation reaction of hispidin. As described above, Mg-ATP, CoA, cysteine, and flavin peroxide are involved in the reactions of Stage 1. These molecules are of many rotatable single bonds, which result in a large number of conformations for each molecule. Indeed, some substituent groups do not qualitatively affect the reaction mechanism. Therefore, in the viable calculations, Mg-ATP, CoA, cysteine, and flavin peroxide were replaced by the simplified molecules acetyl methyl phosphate, methanethiol, and lumiflavin peroxide, respectively, as shown in Figure 2. The detailed simplification processes and reasons are described in the Appendix A. Admittedly, the simplified model is not the actual situation, but it is enough to qualitatively explain the reaction process. The computational results may be quantitatively changed when practical molecules are employed in relevant enzymes.

### 2.2. Computational Method

The Coulomb-attenuated hybrid exchange–correlation functional (CAM-B3LYP) [11], which combines the features of the B3LYP [12,13] with long-range corrections using Hartree–Fock (HF) exchange, and 6−31+G** basis sets were employed in the DFT calculations. Testing of the functionals and the basis sets has been performed previously [3]. This computational level has also been reasonably used in the research of several previous chemiluminescent [14] and bioluminescent [15,16] systems. These calculations involved optimizing the stationary points, analyzing the frequencies, and conducting an intrinsic reaction coordinate (IRC) calculation. To simulate protein environments, solvation effects were considered using the polarized continuum model (PCM) [17,18]. Because the protein solution is complex and cannot be measured accurately, a dielectric constant was selected as ε = 4 by using the information of solid amide and polyamide, whose chemical structures are related to protein [19,20]. All the DFT calculations were performed with the Gaussian 16 program package [21].

## 3. Results and Discussion

### 3.1. Stage 1: Biosynthesis of Fungal Luciferin—From Caffeic Acid to Luciferin

The complete reaction process and predicted reaction mechanism of simplified Stage 1 is summarized in Figure 3, Figure 4, Figure 5 and Figure 6.

#### 3.1.1. Step 1: Biosynthesis of Luciferin Precursors—From Caffeic Acid to Hispidin

The process from caffeic acid to hispidin is essentially a biosynthesis reaction of styryl pyrone compounds, which includes three processes: (1.1) the combination of caffeic acid and CoA in the presence of ATP, (1.2) the two condensations of caffeoyl-CoA and malonyl CoA, and (1.3) the internal esterification of trione intermediates. Process 1.1 can be divided into two stages: (1.1.1) adenylation and (1.1.2) thioesterification. The adenylation reaction means that caffeic acid reacts with Mg-ATP to produce caffeoyl adenosine monophosphate (caffeoyl-AMP) and Mg-PPi. Since Mg-ATP has been simplified to methyl acetyl phosphate in the calculation, this step can also be called phosphorylation. Subsequently, in the thioesterification reaction, caffeoyl-AMP leaves AMP under the attack of CoA and generates caffeoyl-CoA. In CoA ligase, the carboxylic acid substrate is usually stabilized by the surrounding positively charged amino acid residues and exists as an anion. Therefore, the initial state of caffeic acid in the calculation simulation is set to an anion form with a 1-unit charge. CoA is replaced by the simplified model methyl mercaptan in the calculation, and the detailed simplification strategy has been introduced in the calculation details above. The calculation results of Process 1.1 are summarized in Figure 1. The reactant R_1.1.1_ in the phosphorylation process (1.1.1) consists of a caffeic acid anion and a methyl acetyl phosphate anion. During the reaction, the O_a1_ atom with a negative carboxyl part of the caffeic acid anion nucleophilic attacks P_a2_ on the methyl acetophosphate anion, resulting in P_1.1.1_, a product composed of an acetic acid anion and methyl caffeoyl phosphate. The imaginary frequency vibration mode of the transition state TS_1.1.1_ (251i cm^−1^) corresponds to the cooperative stretching vibration of P_a2_-O_a1_ and P_a2_-O_a3_ bonds. The Gibbs free energy barrier of this reaction is 23.0 kcal·mol^−1^. It is reported that histidine near the reaction center in the biological protein environment can stabilize the sulfhydryl anion in the reactant, form a hydrogen bond with the phosphate group, and stabilize the negative charge of the phosphate group, thus assisting its departure process [22]. It can be seen that the absence of histidine will greatly affect the reaction efficiency of thioesterification (1.1.2), and histidine near the reaction center should exist in the formation of the cation. Therefore, in the calculation model of reactant R_1.1.2_ in Process 1.1.2, besides the caffeoyl methyl phosphate and methyl mercaptan anions generated in the previous step, an imidazole cation is introduced to simulate histidine near the reaction center [22,23]. During the reaction, first, S_c1_ nucleophilic attacks C_6_, resulting in the intermediate IM_1.1,2_. Next, C_6_ nucleophilic attacks O_a1_, resulting in the product P_1.1.2_ composed of a thiocaffeoyl methyl ester, a methyl phosphate anion, and imidazole. Two transition states, TS1_1.1.2_ (128i cm^−1^) and TS2_1.1.2_ (139i cm^−1^), were located, and their imaginary frequency vibration modes corresponded to the stretching vibration of the C_6_-S_c1_ and C_6_-O_a1_ bonds, respectively. Imidazole cations form hydrogen bonds with O_a4_ on caffeoyl methyl phosphate and S_c1_ on sulfhydryl, respectively, in the reactant. With the progress of the reaction, the C_6_-S_c1_ bond is formed (10.359 Å → 1.780 Å), the C_6_-O_a1_ bond is broken (1.351 Å → 6.192 Å), and the proton on imidazole cation is gradually transferred to O_a4_. The Gibbs free energy barriers of the thioesterification reaction are 25.9 kcal·mol^−1^ and 24.4 kcal·mol^−1^.

In Process 1.2, caffeoyl-CoA is catalyzed by Hisps to increase the carbon chain through two condensation reactions. Before the condensation reaction, substrate loading and decarboxylation of malonyl CoA should be carried out first. Substrate loading refers to the acyl transfer reaction of caffeoyl-CoA, which transfers the caffeoyl group linked to -SCoA to the sulfur atom of cysteine. After acyl transfer, the structure can still be simplified as a thiocaffeic acid methyl ester, but the sulfur atom is no longer the atom in CoA but the atom in cysteine. To distinguish, in the reactant R_1.2.1_, the sulfur atom of the thiocaffeoylmethyl ester is labeled as S_c2_. The decarboxylation process of malonyl CoA is to form an acetyl CoA anion for the condensation reaction. In the calculation model, the acetyl CoA anion is simplified as a methyl thioacetate anion. Since most of the reaction centers in the biological protein environment also include an important histidine to stabilize the thiol anion of cysteine before and after the reaction [24], it is also necessary to introduce an imidazole cation into the model. Therefore, in the calculation model, the reactant R_1.2.1_ contains three parts: methyl thiocaffeic acid, a methyl thioacetate anion, and an imidazole cation. The simulation results of Process 1.2 are summarized in Figure 2. The mechanism of the first condensation reaction is that C_5_ of methyl thioacetate nucleophilic attacks the C_6_ site of methyl thiocaffeic acid. With the progress of the reaction, a C_5_-C_6_ bond is formed (5.511 Å → 1.527 Å), and the C_6_-S_c2_ bond is broken (1.797 Å → 5.732 Å). The product P_1.2.1_ consists of diketone intermediate, methyl mercaptan, and imidazole. The imaginary frequency vibration mode of the transition state TS_1.2.1_ (229i cm^−1^) connecting the reactant and the product corresponds to the stretching vibration of the C_5_-C_6_ bond. The process of the second condensation reaction is similar to that of the first one. It also needs to be loaded and decarboxylated first so that the reactant R_1.2.2_ consists of a diketone compound, a methyl thioacetate anion, and an imidazole cation. The calculation results show that the imaginary frequency vibration mode of the transition state TS_1.2.2_ (325i cm^−1^) corresponds to the stretching vibration of the C_3_-C_4_ bond, that is, C_3_ nucleophilic attack C_4_. The product contains a trione intermediate, a thiol anion, and an imidazole cation. Under the model system, the Gibbs free energy barriers to be overcome by the two condensation reactions are about 14.3 kcal·mol^−1^ and 8.5 kcal·mol^−1^.

Next, ketosis hispidin can be formed from the intermediate of trione through internal esterification. The simulation results of the lactonization process are shown in Figure 3. The reactant R_1.3.1_ in the model is composed of trione intermediate and imidazole, and imidazole is used to simulate the basic amino acid residues near the reaction center. During the reaction process, first, imidazole attacks the proton H_5′_ at C_5_ the position of the trione intermediate to generate the anion intermediate IM_1.3.1_, and the N atom of the imidazole part forms N-H_5′_ bonds with proton H_5′_ (2.408 Å →1.036 Å). Then, O_1_ nucleophilic attacks C_2_ and O_1_-C_2_ bonds are formed (2.958 Å →1.371 Å), while the C_2_-S bond breaks (1.786 Å →3.461 Å). The product P_1.3.1_ includes an imidazole cation, a methyl mercaptan anion, and ketone-type hispidin. Two transition states, TS1_1.3.1_ (1371i cm^−1^) and TS2_1.3.1_ (173i cm^−1^), are located in the process of internal esterification. The unique imaginary frequency vibration modes correspond to the stretching vibration of the C_5_-H_5′_ bond and the O_1_-C_2_ bond, respectively. The Gibbs free energy barriers of the C_5_ proton leaving and O_1_-C_2_ bonding are 14.3 kcal·mol^−1^ and 7.7 kcal·mol^−1^ respectively. Then, ketone-type hispidin can transfer the protons on C_3_ to O_4′_ by the hydrogen bond network in the protein environment and form enol-type hispidin. In this study, a water bridge built by two water molecules is introduced to simulate the proton transfer process. The reactant R_1.3.2_ and the product P_1.3.2_, respectively, corresponded to ketone-type, enol-type hispidin, and the complex of the two water molecules. The transition state connecting the two structures is TS_1.3.2_ (785i cm^−1^). The Gibbs free energy barrier of isomerization is 7.7 kcal·mol^−1^.

#### 3.1.2. Step 2: Biosynthesis of Luciferin—From Hispidin to Luciferin

As shown in Figure 3, Figure 4, Figure 5 and Figure 6, the enol hispidin synthesized from caffeic acid will be catalyzed by the H3H enzyme to produce luciferin (3-hydroxyhispidin) [7]. The H3H enzyme involved in this process belongs to NAD(P)H-dependent monooxygenase containing flavins. Generally speaking, these enzymes have strict substrate specificity [25]. In the process of hydroxylation, the flavins in the system are first converted into flavin peroxide intermediates through a series of reactions, and then the substrate nucleophilic attacks the flavin peroxide intermediates in the form of anions to produce ketone reaction intermediates, which can be converted into enol hydroxylation products through proton transfer. In this study, the initial form of enol hispidin in the hydroxylation reactant R_1.2_ is set to the anionic form of C_4_-OH deprotonation, and the peroxide intermediate of flavin only retains the isoalloxazine ring, which is simplified to the peroxide intermediate form of photoflavin. The simulation results of the hydroxylation process are shown in Figure 4. The vibration mode of the only imaginary frequency in the transition state TS_2.1_ (607i cm^−1^) is the stretching vibration of the C_3_-O_3′_ bond. Similar to Process 1.3.2, keto 3-hydroxyhispidin can also be converted into the target product enol 3-hydroxyhispidin by isomerization. The reactant R_2.2_ and the product P_2.2_ in the isomerization process (2.2) correspond to the complex of keto-type and enol-type 3-hydroxyhispidin and two water molecules, respectively. The transition state TS_2.2_ (1114i cm^−1^) in the proton transfer process can be located. The Gibbs free energy barriers of hydroxylation and isomerization are 19.8kcal·mol^−1^ and 18.7kcal·mol^−1^, respectively.

The reactions with energy changes in the entire Stage 1 is summarized in Figure 5.

### 3.2. Stage 3: Recovery of Utilization of Oxidation Luciferin—From Oxidized Luciferin to Caffeic Acid

As Figure 1 shows, Stage 2 follows Stage 1 studied above. In Stage 2, under the catalytic oxidation of luciferase, 3-hydroxyhispidin produces oxidized luciferin, namely, caffeoyl pyruvic acid, and emits green light with a wavelength of 530 nm. Stage 2 has been carefully studied [3]. Now, we consider Stage 3 in Figure 1. In Stage 3, caffeoyl pyruvic acid is hydrolyzed into caffeic acid and pyruvic acid under the catalysis of the CPH enzyme. CPH belongs to the fumarylacetoacetate hydrolase (FAH) superfamily [26,27]. These types of enzymes contain highly conserved FAH catalytic domains. Picher et al. reported that the FAHD1 protein, which is a kind of FAH enzyme, can catalyze the hydrolysis of acyl pyruvic acid, and the products correspond to carboxylic acid and pyruvic acid [28,29]. In the reaction process, the substrate first undergoes enol-keto tautomerization to form a diketone intermediate. Then, with the participation of basic amino acid residues in the activation site, it reacts with water molecules to generate a geminal diol anion intermediate. Finally, the corresponding carboxylic acid and pyruvic acid are generated by C-C bond dissociation. The predicted reaction mechanism is shown in Figure 7.

We have calculated the hydrolysis reaction path of caffeoyl pyruvic acid, as shown in Figure 6. First, enol isomers are converted to ketone isomers. During the conversion, the protons on C_4_-OH are transferred to C_5_ by the hydrogen bond network in the system. Similar to Process 1.3.2 and Process 2.2, a water bridge composed of two water molecules is introduced into the calculation model to simulate the proton transfer process. Between enol isomer R_3.1_ and keto isomer P_3.1_, the transition state of proton transfer TS_3.1_ (834i cm^−1^) is located.

In the hydrolysis process (3.2), the reactants include 4,6-deketone isomer of oxidized luciferin, a water molecule, and an imidazole ring, in which the imidazole ring is used to simulate the catalytic base such as histidine near the reaction center. Two transition states (TS1_3.2_ (342i cm^−1^) and TS2_3.2_ (442i cm^−1^)) and an intermediate IM_3.2_ are located between the hydrolysis reactant R_3.2_ and the product P_3.2_. In R_3.2_, the distance between the C_6_ atom and the oxygen atom O_a1_ of the water molecule is about 2.914 Å. First, the oxygen atom O_a1_ of the water molecule attacks the C_6_ position, forming the intermediate IM_3.2_ of the geminal diol. R_3.2_ is connected with IM_3.2_ through the transition state TS1_3.2_, in which the C_6_ atom and O_a1_ atom bonds (2.914 Å→1.454 Å), accompanied by two proton transfer processes, and the proton on the water molecule transfers to the nitrogen atom of imidazole and H_3″_ transfers to O_a_. Then, the C_5_-C_6_ bond dissociates (1.554 Å→4.581 Å), and the product P_3.2_ is obtained. P_3.2_ and IM_3.2_ are connected by the transition state TS2_3.2_, which is also accompanied by two proton transfer processes: the proton connected to O_a1_ is transferred to O_4′_, and H_3″_ is transferred to O_a_. Hydrolysate P_3.2_ consists of a caffeic acid anion, pyruvic acid (enol isomer), and an imidazole cation. The energy barriers of C_6_-O_a1_ bonding and C_5_-C_6_ bond dissociation during hydrolysis are 13.9 kcal·mol^−1^ and 24.7 kcal·mol^−1^, respectively.

In summary, the energy change in the whole Stage 3 is shown in Figure 7.

## 4. Conclusions

The bioluminescent fungi maintain stable BL by continuously synthesizing luciferin. This cycle of synthesizing luciferin is composed of three stages: biosynthesis of luciferin from caffeic acid, luminescence process from luciferin to oxidized luciferin, and regeneration of caffeic acid from oxidized luciferin. In this article, we performed mechanistic studies on Stages 1 and 3. In Step 1 of Stage 1, caffeic acid first undergoes phosphorylation and thioesterification to bond with CoA. The generated caffeoyl-CoA transfers caffeoyl to protein cysteine. An acetyl CoA anion is formed after the decarboxylation of malonyl CoA nucleophilic attacking the C_6_ and C_5_ positions, respectively, and two condensation reactions are carried out to generate trione intermediates. Then, O_a_ nucleophilic attacks C_3_, and a locatonization reaction occurs to generate keto hispidin, and then enol hispidin is obtained through tautomerization. Next, in Step 2 of Stage 1, C_3_ of the hispidin anion nucleophilic attacks the peroxy bond of flavin peroxy intermediate to form keto 3-hydroxyhispidin. Keto 3-hydroxyhispidin transfers the C_3_ proton to O_4′_ to enol 3-hydroxyhispidin, which is a fungal luciferin. In Stage 3, the hydrolysis stage, the tautomerization of enol isomers first occurs. Then, O_a1_ on water molecules nucleophilic attacks C_6_ to form a gemdiol intermediate. The intermediate undergoes a C_5_-C_6_ bond cleavage to obtain pyruvate and caffeic acid. Combined with our previous study on Stage 2 [3], the detailed mechanism of the entire process of the caffeic acid cycle of fungal luminescence has been clearly described. There are so many chemical reactions in all the three stages. Combining Figure 5, Scheme 2 of reference [3], and Figure 7, one can quickly follow the chemical process in fungal BL. We have to say that some of our theoretical studies employed simplified models, and all the calculations were not performed in the actual protein. However, for such a complicated reaction process, these calculations are helpful to understand the cycle of synthesizing luciferin in fungal BL, and will be the basis for more reliable calculations in the future.

## Data Availability

The data presented in this study are available on request from the authors. The data are not publicly available due to privacy.

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
