# Peer review of "Chemistry in Fungal Bioluminescence: Theoretical Studies on Biosynthesis of Luciferin from Caffeic Acid and Regeneration of Caffeic Acid from Oxidized Luciferin"

_jof, 2023, doi:10.3390/jof9030369_

Round 1
Reviewer 1 Report
The presented manuscript describes the development of computational investigations of fungal bioluminescence mechanism. Overall work is neat and sufficient, however I would like to introduce certain changes and comments:
1) The picture of a dragon (ouroboros), though nice and appealing, needs to be redrawn in order to make the captions/text on it more clean and contrast, therefore more readable.
2) The authors should be more accurate with the terms and word usage, for example: "3-hydroxyhispidin" is a one-word term (according to Kotlobay et al 2018), "fungal BL" is better than "fungi BL", I also assume that HispS belongs is a "hispidin synthase", not "synthetase" as belongs to ''polyketide synthase" family, please refer to recent Yampolsky/Sarkisyan group publications for clarification. These are just some examples.
3) For analysis of the potential energy surface (PES) by means of the DFT the authors use the protocol they were using in their previous work (Wang, Liu 2021). I suppose the particular choice of the exchange-correlation functional/basis set in the present paper should be explained not only by referring to the previous work. In Wang, Liu 2021 the authors support their choice of the functional/basis set only by comparison of the results obtained with other type of the functional and larger basis set. I would recommend that the authors provide more general considerations and references on suitability of the chosen functional for investigated types of the molecules and reactions. The similar considerations are relevant for a particular choice of the medium used in accounting for solvatation effects.
/As a general advice, probably for further works, I would suggest to increase the steps of the intrinsic reaction coordinate on shoulders of the PES curves (far from equilibrium/transition states) in sake of the computational resources saving./
4) I would also suggest the improvement of abstract/conclusion parts of the MS. What is the main impact of the investigation and what the possible perspectives? Is there any practical outcome for the researchers working with fungal bioluminescence and its applications? The authors should have any considerations on this topic, they need to introduce them in the paper. Also the conclusion in the current form seem to be scarce and non-specific. Address to the figures in MS and references in the conclusion are not acceptable, it should be able to be read as a separate piece of information.
5) There is also a number misprints, such as line 14 "fuNgal", "in vivo" (italics), excessive "methanthiol," in line 82 etc.
Overall impression on the MS is that it has to be improved to make it more readable, neat and relevant, but minor changes are actually required.
Reviewer 2 Report
The manuscript described the theoretical mechanism of the entire process of caffeic acid cycle of fungal bioluminescence. Totally, the manuscript was well written by step-by-step. Broad reader will be interesting in this manuscript. However, it is not so easy to understand the detailed discussion because the overview of total mechanism in “Scheme 3” is complicated in comparison with, for example, schemes 1 or 6. It is also not so easy to check the blue and black compounds. The authors should consider “Scheme 3” and, for example, may separate each step for “Scheme 3-1, 3,,,”.
I have other minor comments as follows:
1) Sentence in line 128 needs suitable reference.
2) Sentence in line 227 needs suitable reference.
3) Please check line 28 and 82.
Reviewer 3 Report
In their manuscript, the authors investigated the mechanism of stages 1 and 3 of the caffeic acid cycle in fungal bioluminescence using a DFT methodology. While the manuscript is of potential interest to the broad readership of the Journal of Fungi, the following issues need to be addressed before publication:
Major Issues:
1. While I see the potential importance of this study, it is also important to articulate the importance of the study to the readers why they have used only one DFT functional, CAM-B3LYP. Because, the choice of method is very crucial, and in particular when using DFT, the choice of functional is very critical. So, using a particular functional for exploring a reaction mechanism is not a good idea (unless there is a benchmark study on the similar/same system). I would suggest author could check (not the whole study, but for a few steps, such as the phosphorylation and thioesterification process ) if using a different functional can improve (or differ) the findings of this study.
2. The authors used 6-31+G** basis set for the entire study and didn’t check with any other basis set, to see if the basis set will have any effect on their results. Can the author comment on it?
3. The discussion is based on simplified molecules (of original reactants) concerning enzymatic reactions. It is understandable that due to the unavailability of crystallographic data on the relevant enzymes, it is not possible to take into account the effect of the enzymatic environment. However, since most of the studied reactions are undergoes nucleophilic attack, steric factors would play a critical role in these reactions. Therefore, authors should comment on choosing their “simplified molecules”. How would the bulkiness of the “simplified molecules” affect the current result if we were to use a bulkier version of actual reactants?
4. The “Conclusion” needs to be revised. A conclusion should have important pieces of information coming from the author’s calculation. The current material in the “Conclusion” basically has similar information to what is already written in the final paragraph of the “Introduction”.
Minor Issues:
1. Page 3, line 82, “methanethiol” is written twice, should be once.
2. The figures also need to be reduced or combined. Figures 1 and 2 can be combined. Figures 3 and 4 can also be combined. Some Figures can go to SI. Please try to do this to improve the impact and readability of the paper.

Round 2
Reviewer 3 Report
I have carefully evaluated your manuscript and the reviewers’ reports, and the reports give satisfactory answers to previous comments. Thus, I recommend this article be published in its current form.